# Prediction of risk of acquiring urinary tract infection during hospital stay based on machine-learning: A retrospective cohort study

**Jens Kjølseth Møller**[1]*, **Martin Sørensen**[2], **Christian Hardahl**[3]

**1** Department of Clinical Microbiology, Lillebaelt Hospital, University Hospital of Southern Denmark, Vejle, Denmark, **2** SAS Institute A/S, Copenhagen, Denmark, **3** SAS Institute A/S, Aarhus, Denmark

\* jens.kjoelseth.moeller@rsyd.dk

## Abstract

### Background

Healthcare associated infections (HAI) are a major burden for the healthcare system and associated with prolonged hospital stay, increased morbidity, mortality and costs. Healthcare associated urinary tract infections (HA-UTI) accounts for about 20–30% of all HAI's, and with the emergence of multi-resistant urinary tract pathogens, the total burden of HA-UTI will most likely increase.

### Objective

The aim of the current study was to develop two predictive models, using data from the index admission as well as historic data on a patient, to predict the development of UTI at the time of entry to the hospital and after 48 hours of admission (HA-UTI). The ultimate goal is to predict the individual patient risk of acquiring HA-UTI before it occurs so that health care professionals may take proper actions to prevent it.

### Methods

Retrospective cohort analysis of approx. 300 000 adult admissions in a Danish region was performed. We developed models for UTI prediction with five machine-learning algorithms using demographic information, laboratory results, data on antibiotic treatment, past medical history (ICD10 codes), and clinical data by transformation of unstructured narrative text in Electronic Medical Records to structured data by Natural Language Processing.

### Results

The five machine-learning algorithms have been evaluated by the performance measures average squared error, cumulative lift, and area under the curve (ROC-index). The algorithms had an area under the curve (ROC-index) ranging from 0.82 to 0.84 for the entry model (T = 0 hours after admission) and from 0.71 to 0.77 for the HA-UTI model (T = 48 hours after admission).

**Data Availability Statement:** The data underlying this study are publicly available at doi:10.5061/dryad.6djh9w107.

**Funding:** This study received funding from the Region of Southern Denmark Research Foundation, 17/15659, awarded to JKM.

**Competing interests:** The authors have declared that no competing interests exist.

## Conclusion

The study is proof of concept that it is possible to create machine-learning models that can serve as early warning systems to predict patients at risk of acquiring urinary tract infections during admission. The entry model and the HA-UTI models perform with a high ROC-index indicating a sufficient sensitivity and specificity, which may make both models instrumental in individualized prevention of UTI in hospitalized patients. The favored machine-learning methodology is Decision Trees to ensure the most transparent results and to increase clinical understanding and implementation of the models.

## 1. Introduction

Healthcare associated infections (HAI) are a major burden for the healthcare system and are associated with prolonged hospital stay, increased morbidity, mortality and costs. Healthcare associated urinary tract infections (HA-UTI) accounts for about 20–30% of all HAI [1–7] and with the emergence of multi-resistant urinary tract pathogens, the total burden of HA-UTI will most likely increase. The present study is a part of a larger study where the primary aim is to create computerized algorithms for continual surveillance of the incidence of various types of HAI [6, 7].

A natural extension to the aim of automated electronic surveillance of HAI is to create a model predicting the risk of an admitted patient developing HA-UTI. The adoption of Electronic Medical Records (EMRs) in many hospitals and advances in data science creates an opportunity to develop big data-driven machine-learning algorithms [8, 9]. Trained and evaluated on patient cases derived from automated registration and surveillance of hospital-acquired infections with the purpose of predicting individuals admitted to hospital with a greater risk of developing serious infections such as surgical site infections, bacteremia, and HAI-UTI during admission [10–13]. A real-time early warning system for monitoring inpatient disease risk should reflect underlying clinical causality and thereby enable caregivers to provide clinical intervention or treatment, e.g., in case of inpatient mortality risk [14]. A successful risk predictive model of HA-UTI may identify potential risk patients allowing the care providers to take proper measures to lower the risk.

A number of studies have identified several risk factors for developing HA-UTI. King et al. presents a systematic review, calculating Population Attributable Risk (PAR) for risk factors included in 23 papers. Independent risk factors with a high and significant PAR include urinary catheter, stroke, female sex, and hypertension [15]. Redder et al. also identified admission with CA-UTI, diseases in the genitourinary system and diseases of the nervous system as significant factors influencing the odds of developing a HA-UTI [16].

The overarching aim of our study is to create a real-time early warning system for individualized prevention of UTI in hospitalized patients using structured and unstructured data recorded at the hospital for predicting patient risk of acquiring HA-UTI. The primary aim, therefore, was to train, validate, and compare predictive models for UTI in large independent groups of admitted patients using machine-learning algorithms on data from an existing daily updated central data warehouse containing a copy of the electronic medical record (EMR) dataset of our regional somatic hospital trusts. The secondary aim was to develop and compare various predictive models that predict the probability of acquiring 1) any UTI (CA-UTI or

HA-UTI) at the entry time of admission (= Entry Model), and 2) a HA-UTI after 48 hours of admission (= HA-UTI Model).

## 2. Materials and methods

### 2.1 Study design and population

The study is a retrospective cohort analysis of consecutive patient admissions in a Danish region over a 16 months period beginning January 2017. Data were obtained from the four public somatic hospital trusts in the Region of Southern Denmark comprising 301,932 adult admissions. The public health system is tax-financed and consequently free of charge for the individual patient and the few small private hospitals in Denmark only cover a negligible number of admissions. According to Danish legislation, no approval from an ethics committee or consent from participants is required for registry-based studies. The four regional hospital trust authorized the use of de-identified regional data for this research project. All personal information was masked during the process of analysis and publication. The study was approved by the Danish Data Protection Agency (journal number 2008-58-0035). According to EU regulation 2016/679, a Data Protection Impact Assessment (DPIA) has been performed. The DPIA describe the nature, scope, context and purpose of the extensive profiling or automated decision-making to make significant decisions about our patients; identify and assess risks to individuals; and identify any additional measures to mitigate those risks. All data were stored on encrypted computers and servers. The dataset used is available in the Dryad Digital Repository (doi:10.5061/dryad.6djh9w107). Data files contain no sensitive or personally identifiable information.

### 2.2 Data source

Data are extracted from a regional data warehouse containing 1) a copy of EMR notes from the four hospital trusts in the Region of Southern Denmark, which covers a population of 1.2 million inhabitants; 2) administrative data on all hospital admissions; 3) the positive culture results from the four clinical microbiology laboratories in the region (Fig 1).

The central regional copy of EMR consists of both structured and unstructured data. Data from admission to intensive care units are not included. Data include information about all patient and their admission including data on diagnoses, procedures, administered drugs, prescribed drugs, and microbiology and biochemistry lab results (Fig 1). All Danish residents have from cradle to grave the same unique civil registration number used for all health contacts in Denmark, which enables linkage between the various public healthcare registries and the construction of the medical history of a patient [17].

### 2.3 Data set creation and definitions

The development data set on adult admission is extracted from the regional data warehouse. Data types include dates and place of admission, demographics, historical diagnosis codes, data from an automatic electronic infection registry, trigger based text mining from the electronic medical records in hospital (EMR) [18, 19]. Risk factor variables chosen were based upon review of the literature [15] and our previous study analyzing risk factors for urinary tract infection [16, 20]. The target variable to be used in the predictive models is any urinary tract infection (UTI) detected during admission. Urinary tract infection is predominantly diagnosed based on a combination of clinical features, culture of significant amounts of bacterial pathogens in urine, and relevant antimicrobial treatment [1, 6]. Information about UTIs is copied from the automated electronic hospital infection monitoring system HAIR used in the

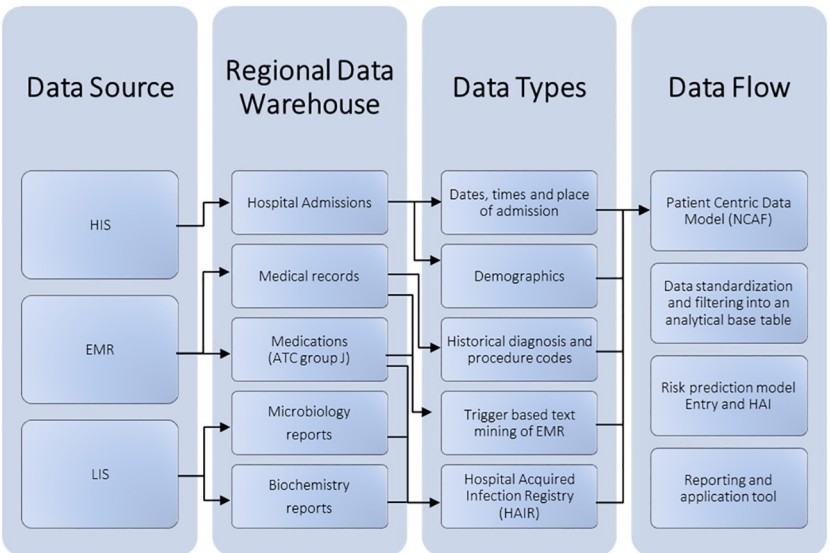

**Fig 1. Data landscape.** HIS: Hospital Information System, EMR: Electronic Medical Record, LIS: Microbiology Information System.

Region of Southern Denmark [6, 7, 16, 20]. All admitted patients were automatically classified by HAIR as having 1) a community-acquired urinary tract infection (CA-UTI), 2) a Hospital-acquired urinary tract infection (HA-UTI), or 3) no UTI based on data from the regional data warehouse and algorithms previously described. The algorithms for UTI are based on modified CDC definitions, S1 Table [6, 7, 21]. A HA-UTI is defined as a urinary tract infection discovered in a hospitalized patient, who at the time of discovery has been hospitalized more than 48 hours. A CA-UTI is defined as a urinary tract infection discovered in a hospitalized patient, who at the time of discovery has been hospitalized less than 48 hours. Control cases comprise the admissions without UTI from the same period.

The overall study cohort was composed from admissions between January 2017 and May 2018 in a forward consecutive manner selecting all patients ≥18 years of age admitted to any of the medical or surgical departments of the somatic hospitals in the Region of Southern Denmark. Seasonal changes of UTI infection patterns may occur but the present use of more than one year of data should be sufficient to generalize for scoring new cases. From a modelling perspective, we are able to create models that perform well on new data. All data between January 2017 and April 2018 were randomly divided (60%/40%) into independent partition sets for training (creation of the models), validation (model selection). The two sets of random samples were stratified by the target variable. A third time-independent test data set (test of model performance on new/unknown data) was created from admissions in May 2018. The characteristics of the study population are shown in Table 1.

In case there were several episodes that fulfilled the criteria of either HA-UTI or CA-UTI during a single care episode (admission), a window of 7 days after the first positive event was applied where no new events could be registered unless a new, non-matching pathogen was detected in a urine sample of the patient.

## 2.4 Data pre-processing

Categorical variables were included in the models in their current categorical form. All continuous variables were placed into 2 or more discrete categories, e.g., no fever (= 0) or fever (= 1;

**Table 1. Selected characteristics of the study population.**

| Characteristic | Training (n, %) | Evaluation (n, %) | Test (n, %) |
|---|---|---|---|
| | N = 170,533 | N = 113,760 | N = 17,533 |
| Time period | Jan 2017 –Apr 2018 | Jan 2017 –Apr 2018 | May 2018 |
| Target Entry model (CA-UTI or HA-UTI) | 6.2% | 6.2% | 6.4% |
| Target HAI model (HA-UTI) | 2.4% | 2.4% | 2.6% |
| Age (years, mean + SD) | 51.6 (27.3) | 51.6 (27.4) | 52.0 (27.4) |
| Sex (n, % male) | 46.1% | 46.2% | 46.4% |
| Readmission rate | 8.5% | 8.5% | 8.9% |
| ICD-10 diabetes, at admission | 3.2% | 3.2% | 3.8% |
| ICD-10 urinary retention, at admission | 0.7% | 0.6% | 0.9% |
| ICD-10 kidney (DN18), at admission | 1.4% | 1.4% | 1.9% |
| ICD-10 stroke, at admission | 1.9% | 1.9% | 2.4% |
| ICD-10 hypertension, at admission | 5.3% | 5.3% | 6.8% |
| ICD-10 unconsciousness, at admission | 1.0% | 1.1% | 1.4% |
| ICD-10 COPD[1], at admission | 3.0% | 3.0% | 3.5% |
| ICD-10 atrial fibrillation, at admission | 0.4% | 0.4% | 0.5% |
| ICD-10 CVC[2] start, at admission | 1.1% | 1.1% | 1.8% |
| ICD-10 CVC[2] end, at admission | 0.1% | 0.0% | 0.2% |
| OPCS[3] bladder scan, at admission | 2.3% | 2.2% | 3.5% |
| ICD-10 neuro disease, at admission | 2.7% | 2.7% | 3.5% |
| Previous CA- UTI[4] | 3.75% | 3.75% | 5.6% |
| Previous HA- UTI[4] | 2.2% | 2.4% | 3.7% |
| CA-UTI during current admission[4] | 4.2% | 4.2% | 4.4% |
| Trigger: Previous IUC[5] | 7.8% | 7.7% | 12.3% |
| Trigger: IUC during admission | 7.2% | 7.1% | 7.1% |
| Trigger: Urine retention | 0.2% | 0.2% | 0.3% |
| Trigger: Fever ($\geq$ 38˚C) | 9.5% | 9.5% | 9.0% |

CA-UTI: Community Acquired UTI; HA-UTI: Hospital Acquired UTI.

[1]Chronic obstructive pulmonary disease.

[2]Central venous catheter.

[3]OPCS: operation and procedure codes.

[4]UTIs determined by HAIR, data before January 2017 not included.

[5]Presence of indwelling Urinary Catheter (IUC) before January 2017 not included.

Trigger: text mining of clinical narrative text in EMR.

defined by temperature $\geq$ 38˚C). Observational data were only part of the HAI model (Table 2). Missing values were included and treated as categorical values (= 0) within the models.

**2.4.1 Variables.** The time spent at the hospital is assumed to affect the risk of developing a HA-UTI. Only factors up until the day of hospitalization are considered to add to the risk of developing a CA-UTI, Fig 2 and Table 2 shows the in-put variables used in the entry and the HA-UTI model. In-put variables were primarily chosen to reflect possible underlying clinical causality, e.g., documented instrumentation of the urinary tract, urine retention diagnosed, presence of Indwelling Urinary Catheter, or previous UTIs detected.

**2.4.2 Text mining and trigger construction.** Text mining (TM) was performed by compiling data on clinical UTI signs, symptoms, and findings documented in unstructured, clinical narrative text from the regional EMR data warehouse using SAS® Content Categorization

**Table 2. Variables used in the entry model and the HAI model.**

| Entry Model[1] | HA-UTI Model | |
|---|---|---|
| **Registry data** | **Registry data** | **Observational data**[2] |
| • Gender | • Gender | • Fever > 38 |
| • Age at admission | • Age at admission | • Dysuria |
| • Readmission (yes/no, previous admission within 30 days) | • Readmission (yes/no, previous admission within 30 days) | • Frequency |
| | | • Urgency |
| • ICD10 codes at admission (diabetes, urinary retention, kidney disease, stroke, hypertension, unconsciousness, Chronic obstructive pulmonary disease) | • ICD10 codes up until HAI or discharge date (diabetes, urinary retention, kidney disease, stroke, hypertension, unconsciousness, Chronic obstructive pulmonary disease) | • Suprapubic tenderness |
| • Prior UTI | • CAI during this admission (from admission time until <48 hours after) | • IUC during admission |
| • Previous IUC | | |

[1]Data available at admission time from previous admissions.

[2]Data obtained by text mining (trigger) from EMR notes during admission and converted to structured variables (yes/no).

(SAS Institute Inc., Cary, USA) [22]. Natural language processing (NLP) algorithms were programmed in the TM software to distinguish between relevant and irrelevant information in the clinical narrative text.

Triggers were based on medically appropriate single or multiple Danish, Latin or English words and phrases, as they appear in the clinical narrative text [18]. Each UTI trigger is shown in Tables 1 and 2. A specific UTI trigger dictionary was created and incorporated into the TM software. Note that every conceivable abbreviation and spelling including misspellings typically found in clinical narrative text were included in the dictionary to account for commonly known, but poorly documented words and phrases in the clinical narrative text. Once the TM software finds a trigger in the clinical narrative text, its context is critically analyzed. Depending on the presence of negations, such as no, not, none, non-, etc., and the position of a trigger and the negation in the sentence, and on the maximum number of words between a trigger or a negation, the trigger is either accepted and classified with its reference number or ignored. Eriksson and colleagues [23] describe the principles of context analysis in detail in their study. Triggers found (trigger reference number, time and date of the clinical narrative text in the EMR) are assigned to the correct patient using the unique Danish civil registration number.

**2.4.3 Target (UTI).** HA-UTI according to the definition, yes (= 1) no (= 0), is the target variable of interest in the HA-UTI Model. Any UTI (CA-UTI or HA-UTI) according to the HAIR criteria used (S1 Table), yes (= 1) no (= 0), is the target variable of interest in the Entry Model.

**2.4.4 Definitions used in classification of data.** A permanent catheter is an indwelling urinary catheter (IUC) defined as a drainage tube that is inserted into the urinary bladder through the urethra, is left in place, and is connected to a closed collection system. Exclusion examples: suprapubic, intermittent (including self-intermittent), external catheter (condom), urostomy, nephrostomy etc.

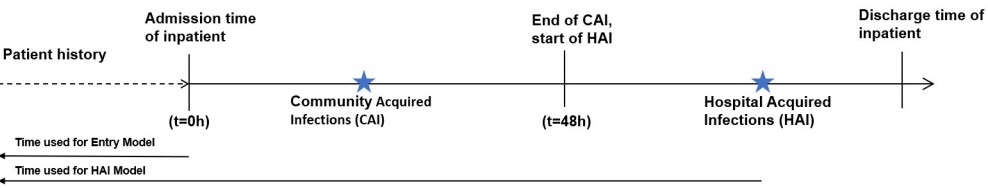

**Fig 2. Illustration of the time lines of the two predictive models; the entry model and the HA-UTI model.**

Time (date) of onset of infection will be considered equivalent as the first occurring of 1) Time of first signs or symptoms of a UTI; 2) Time when urine sample for culture is taken; 3) Time of start of UTI relevant or UTI specific antimicrobial treatment.

Hospitalization period is defined as the time between admission and discharge from any regional hospital. If a discharge is followed by an admission within 24 hours this will be considered as part of the same hospitalization period.

## 2.5 Predictive modelling approach

The primary goal of our modelling approach is to find patterns/relations in data between the dependent variable (target, e.g. HA-UTI) and the independent variables (predictor variables or risk factors). To predict the risk of occurrence of a UTI during admission, we developed two models for UTI prediction 1) an entry model (t = 0), and 2) the HA-UTI model (t = 48) using five different machine-learning algorithms: Neural Networks, Gradient Boosting, Regression, Decision Tree 3 Way Split, and Decision Tree. All models are on target HA-UTI or target Any UTI (HA-UTI or CA-UTI), yes (= 1) versus no (= 0).

The models applied include models that are easily interpretable, such as Decision Trees and Regression models, but also models with a more complex interpretation, such as Gradient Boosting and Neural Networks. Using both tree-based models and regression based models and both simple and more complex models helps us finding the model type that is best at finding the underlying pattern in the data. The data has very little missing values, which is why no imputation has been used.

The models are trained on the training partition of the Development data set. We have trained several different model complexities ranging from low to high model complexity for every model type. The training process stops when one of a set of model specifications / stopping criteria has been met. The criteria used are the following: Neural Network, (number of hidden layers: 1, number of hidden units: 10, max. iterations: 50, max. time: 4 hours); Gradient Boosting, (number of iterations: 50, max. branch: 2, max. depth: 3, min. categorical size: 5, min. leaf fraction: 0.001); Regression, model selection based on the stepwise selection method; Decision Tree, (max. tree depth: 6, min. leaf size: 5, min. categorical size: 5, max. branch: 2); Decision Tree 3 Way Split, (same criteria as the Decision Tree Model, but with max. branch: 3). The primary assessment metric used is average squared error on validation data.

The validation partition of the Development data set is used to assess all models created with the training data. The development data set has been partitioned into 60% for training and 40% for validation. Reasons for this balanced partition choice are that the data set size is big and the high risk patient groups can contain few cases. Assessment and comparisons of models will be based on average squared error, ROC-index, and cumulative Lift.

The aim of predictive modelling is to find the model complexity that result in the lowest error rate on the validation data. We want to find the model complexity on the training data that best generalizes (best fit) on new data (validation data).

SAS Enterprise Miner 14.3 has been used for the model development. Variable selection has been done using automate variable selection procedures, such as Stepwise Selection for Regression Models and Log-Worth for Decision Trees. For Neural Networks and Gradient Boosting, auto-tuning options have been used.

## 2.6 Calculation of individual risk score for acquiring a UTI at admission

After evaluating all relevant candidate models, we select one model as the champion model based on model performance on the validation data. The algorithm used for calculation of an individual risk score per patient depends on the specific machine-learning model that is

selected as the champion model for the entry and the HA-UTI model, respectively. A secondary goal of our modelling approach is that the favored machine-learning model should present transparent results to increase clinical understanding of the model. We choose to focus primarily on the entry-model in the presentation and evaluation of results in the present study.

For a Decision Tree model (champion model), the calculation of an individual risk score is based on the probability of acquiring a UTI estimated for each end node of the tree (the end nodes are the branches of a tree where no further branches are created). If a tree has 36 end nodes then 36 probability groups are created. The probability group that will be attached to each patient depends on the characteristics (data values) of the patient. Example (S2 Fig): IF the patient is below age 74 AND the patient has at least two previous community acquired infections THEN the probability of acquiring any UTI is about 6 times higher than the basic risk (37.6% versus 6.3%). All patients will have a probability group attached depending on their characteristics.

## 3. Results

Over the study period of 16 months, 284,293 unique admission IDs were included and constituted a training and validation data set. In total, 17,768 cases (6.3%) had a UTI (CA-UTI and HA-UTI) defined by HAIR. No difference in the distribution of UTIs was seen in the training and validation data set (Table 1). In the one-month independent test data set, a slightly higher percentage (6.6%) of admissions had a UTI registered (Table 1). The median age for the patients in the training, validation, and test data set was 51.6 (SD = 27.3), 51.6 (SD = 27.4), and 52.0 (SD = 27.4), respectively (Table 1). Male patients accounted for 46% in all groups. Additional patient characteristics are shown in Table 1.

Classification results for the machine-learning models used on the entry model and the HA-UTI model are presented in Fig 3 and S2 Table.

For the entry model, the ROC index (area under the curve) ranged for the Validation data set from 0.812 (Decision Tree) to 0.833 (Neural Network), and for the Test data set from 0.818 (Decision Tree) to 0.841 (Neural Network). No significant difference was thus observed between the five machine-learning models assessing the entry model (S2 Table).

For the HA-UTI model, the machine-learning models had a ROC index ranging from 0.736 (Decision Tree) to 0.777 (Neural Network) for the Validation data set, and a ROC index ranging from 0.709 to 0.770 for the Test data set (S2 Table).

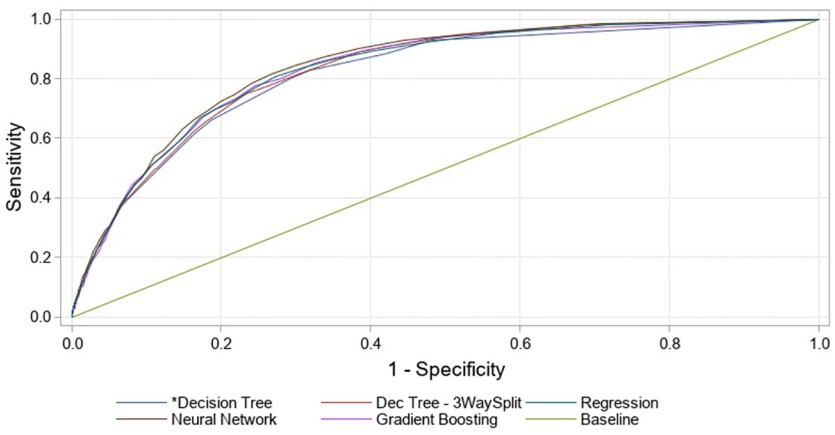

**Fig 3. Receiver Operating Characteristic (ROC) curves for the machine learning models.**

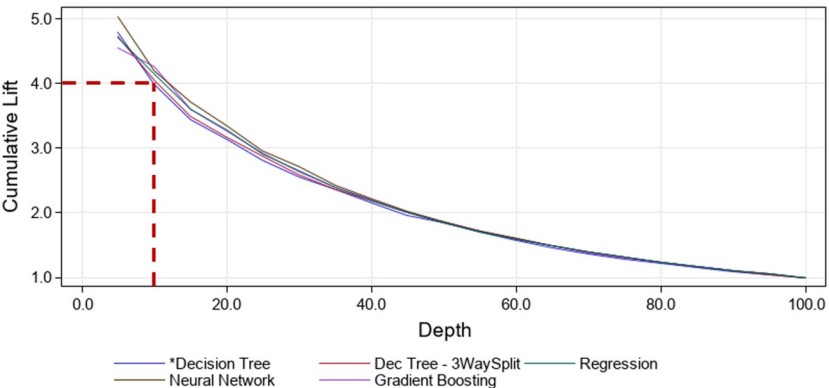

**Fig 4. The cumulative lift for the entry model.**

In Fig 4 is shown a cumulative lift for the entry model of about four times for the top 10% of the admissions (test data set) with the highest probability of patients acquiring a UTI for all machine-learning models employed. The predicted probability of UTIs were sorted and plotted in descending order. A cumulative lift of four times means that among the 10% of patients with the highest probability we will find four times as many admissions with a UTI than we would have from a random sample of all admissions. The corresponding cumulative lift (top 10% of admissions) for the HA-UTI model was approximately 2.4 (S3 Fig).

The variable importance table (Table 3) shows the predictor variables that the Decision Tree machine-learning model used for the entry-model. For the HA-UTI model a similar variable importance table (S3 Table) was created. As indicated by the number of splitting rules for the various input variables in the entry-model (Table 3), the univariate and multivariate analyses of the risk factors included in the study show that admission type (acute admission), age at admission, sex, and past UTI are impactful predictive features in the Entry-model (Table 4).

A Dashboard for operationalizing of the results was developed for presentation of patients and their corresponding risk in a web based reporting tool, S4 Fig (overview). S5 Fig presents an example of the values for the most important variables in the Entry Model for a selected patient. A preliminary prospective study (data shown in S1 Appendix) was executed using the entry model at Department of emergency medicine at Vejle Hospital in the Region of Southern Denmark. For patients admitted more than 48 hours, a urinary tract infection (HA-UTI or CA-UTI) was detected during admission in 51.9% (111/231) of patients with a risk ratio $\geq$0.18 at entry, in 26.0% (120/461) with a risk ratio between 0.18 and 0.08, and in 16.3% (25/153) with a risk ratio $\leq$ 0.08 at entry, respectively.

Thus 72% (185/256) of patients acquiring a HA-UTI during admission had a risk score of $\geq$0.15 at the time of admission (data shown in S1 Appendix).

## 4. Discussion

We developed two models for UTI prediction, an entry model (time of admission) and a HAI model predicting the risk of acquiring a UTI after 48 hours. Both models achieved a high ROC-index for classifying patients at risk of acquiring a UTI during hospitalization. The ultimate aim of the study is to provide staff at clinical departments with an online application integrated in the hospital management system to identify patients at high risk of acquiring a UTI during admission allowing caregivers to provide clinical interventions before the UTI may occur.

**Table 3. The input (predictor) variables used in the decision tree machine-learning model for the UTI prediction at the time of admission (entry-model).**

| Input variable (at admission) | Number of splitting rules | Training Importance | Validation Importance | Ratio of validation to training importance |
|---|---|---|---|---|
| Age | 5 | 1.0000 | 1.0000 | 1.0000 |
| Previous CA-UTI | 5 | 0.8156 | 0.8189 | 1.0042 |
| Admitted_org_id_text | 4 | 0.6104 | 0.5786 | 0.9479 |
| Admission_type | 4 | 0.2578 | 0.2859 | 1.1087 |
| Previous HA-UTI | 5 | 0.2223 | 0.2287 | 1.0287 |
| Sex | 3 | 0.1693 | 0.1615 | 0.9542 |
| Previous IUC[1] | 3 | 0.1635 | 0.0816 | 0.4994 |
| ICD-10 Urinary retention | 1 | 0.1349 | 0.0405 | 0.3001 |
| Readmission | 1 | 0.1107 | 0.0994 | 0.8978 |
| Admission_hospital_text | 2 | 0.1004 | 0.0600 | 0.5978 |
| ICD-10 COPD[2] | 1 | 0.0675 | 0.0705 | 1.0437 |
| ICD-10 Neurological disease | 1 | 0.0381 | 0.0270 | 0.7080 |
| OPCS CVC[3] time of removal | 0 | 0.0000 | 0.0000 | |
| OPCS CVC[2] replacement | 0 | 0.0000 | 0.0000 | |
| OPCS CVC[3] time of insertion | 0 | 0.0000 | 0.0000 | |
| Referral diagnosis | 0 | 0.0000 | 0.0000 | |
| ICD-10 Hypertension | 0 | 0.0000 | 0.0000 | |
| ICD-10 Unresolved spinal injury | 0 | 0.0000 | 0.0000 | |
| ICD-10 Diabetes | 0 | 0.0000 | 0.0000 | |
| ICD-10 Unconsciousness | 0 | 0.0000 | 0.0000 | |
| ICD-10 Chronic kidney disease | 0 | 0.0000 | 0.0000 | |
| ICD-10 Stroke | 0 | 0.0000 | 0.0000 | |
| ICD-10 Atrial fibrillation | 0 | 0.0000 | 0.0000 | |
| OPCS Bladder scan | 0 | 0.0000 | 0.0000 | |

CA-UTI: Community Acquired UTI; HA-UTI: Hospital Acquired UTI.

[1]IUC: Indwelling urinary catheter.

[2]Chronic obstructive pulmonary disease.

[3]Central venous catheter.

OPCS: operation and procedure codes.

Therefore, we developed a simple early warning system for clinical departments providing not only an overall patient specific risk score but also showing the potential of implementing preventive measures to counteract or modify individual risk factors of developing a UTI, S4 and S5 Figs. A preliminary pilot study using the entry model prospectively at a single clinical department shows that close to three quarters of patients acquiring a HA-UTI during admission had a risk score of $\geq 0.15$ at the time of admission.

It is important that an early warning system for monitoring inpatient disease risk factors not only reflects underlying clinical causality but also provides useful information in real-time. This enables caregivers to initiate clinical intervention or treatment to prevent or modify the predicted disease event as early as possible [14]. A good model and user interface for warning the healthcare staff will mean less time needed to examine the EMR of a patient, and is generally based on more experience/learning than a single healthcare provider can obtain through his/her lifetime. We did not observe major differences between the five examined machine-learning methodologies by the default settings employed (ROC curves, Fig 3). The Decision Tree model was chosen as the proposed Champion Model for the machine-learning

**Table 4. Univariate and multivariate analysis of the impactful predictive feature in the entry-model.**

| Predictive feature | Odds Ratio | Analysis of Maximum Likelihood Estimates | | | |
| --- | --- | --- | --- | --- | --- |
| | | DF | Estimate | Standard Error | Wald Chi-Square |
| ADMISSION_TYPE (Acute vs Planned) | 2.286 | 1 | 0.4134 | 0.0210 | 388.04 |
| Sex (Female vs Male) | 1.620 | 1 | 0.2412 | 0.0112 | 459.85 |
| ICD-10 COPD[1], admission (0 vs 1) | 1.244 | 1 | 0.1093 | 0.0236 | 21.38 |
| ICD-10 Neuro Disease, admission (0 vs 1) | 0.725 | 1 | -0.1606 | 0.0255 | 39.79 |
| ICD-10 Diabetes, admission (0 vs 1) | 0.789 | 1 | -0.1184 | 0.0226 | 27.56 |
| ICD-10 Urinary retention, admission (0 vs 1) | 0.496 | 1 | -0.3506 | 0.0382 | 84.10 |
| Age_admission | 1.040 | 1 | 0.0395 | 0.000745 | 2811.38 |
| Previous CAI-UTI (0 vs 1) | 1.748 | 1 | 0.5586 | 0.0238 | 550.17 |
| Previous HAI-UTI (0 vs 1) | 1.153 | 1 | 0.1421 | 0.0363 | 15.29 |
| Previous IUC[2] (0 vs 1) | 1.178 | 1 | 0.1642 | 0.0186 | 77.66 |
| Readmission (0 vs 1) | 1.167 | 1 | 0.0772 | 0.0185 | 17.47 |

CA-UTI: Community Acquired UTI; HA-UTI: Hospital Acquired UTI.

[1]Chronic obstructive pulmonary disease.

[2]IUC: Indwelling urinary catheter.

methodology to be used because of a high ROC-index. The Decision Tree model ensures transparent results and clinical understanding of the two predictive models for UTI as illustrated in S2 Fig. The tools used for this solution are all based on a SAS® Platform utilizing in-memory analytics tools (S2 Appendix).

Good data quality is of paramount importance for the content categorization for text analytics. Our trigger solution is based on our previous experiences with text mining electronic medical records to identify hospital adverse events [18, 19]. Assuming that manual review of electronic medical records gives the truth (the reference method), we found that the true positive fraction using text mining for identifying records of patients with pressure ulcers (all grades) was 70% (95-CI: 55% to 82%) and the true negative fraction was 95% (95-CI: 93% to 97%) [18]. Updating the machine-learning models over time is equally important. Target data in our two UTI predictive models derive from the automatic classification of UTIs in HA-UTI or CA-UTI by the Regional HAIR system [7], and are based on real-time data automatically transferred from the departments of clinical microbiology. With the current data warehouse setup, the Entry Model is updated with new admitted patients hourly. Estimated time from a patient is registered in the database to a prediction is returned to the clinical system is only a matter of seconds for one patient. Depending on the right integrations to the source system it would be possible to do this in near real-time. Changes in classification codes of sample material and pathogens cultured should currently be reflected in the UTI algorithms of HAIR [6, 7]. Recalibration of the UTI predictive models may thus be needed over time and ROC curves recalculated with fixed time intervals. In case new ROC curves indicate a drop in performance of the models, steps may be taken to adjust the models according to conclusions of root-cause-analyses performed. Notably, patients moving to health care trusts from other health-care systems with their past medical history being kept outside the regional EMR may lead to biased associations and impact prediction models. Flagging these patients in the database and excluding them from calculations of the predictive models for a period may counteract such bias.

The strength of the study is the use of an extensive material of patient admissions from a multi-site hospital environment in a Danish region in the form of three independent sets of data to train, validate and test the two predictive UTI models created. Furthermore, that the

target variables (previous community or hospital acquired urinary tract infections) were derived from patient specific surveillance data based on automated computer based algorithms (HAIR) previously validated [6, 7, 16, 20].

It is also an important feature of the present automated computer-based prediction of risk of acquiring UTI during admission that it places no data collection burden on health care staff. The use of objective machine-learning based algorithms including all admitted patients is a major advantage compared to manual and often subjective evaluations of smaller number of randomly selected patient reading through the EMR for individual UTI risk factors. Periodically evaluations of false positive or false negative prediction scores for UTI may enable a dynamic adjustment of the models by analyzing and correcting potential flaws in the predictive models.

It is a limitation that our two predictive models are currently based on EMR data in the regional database, which are only updated once daily from the hospital EMR. Use of real-time data in the risk scoring system could enhance the performance of both models and present more timely predictive scores for risk of acquiring UTI. The percentage of HA-UTI was 2.4% versus 3.8% community-acquired UTIs (CA-UTI) in both the training and the evaluation data set. This may constitute a target variable balance and less risk of a bias towards factors favoring the prediction of either CA-UTI or HA-UTI. The slightly higher percentage of HA-UTI (2.6%) in the test-group of admissions following the months of collecting training and evaluation data is probably caused by a smaller data set (greater statistical uncertainty). It may also be influenced by a data observation period of 16 months before the patient admissions compared to no data available from the period before the collection of the training and evaluation data set. It is important that the driving features in the machine learning models represent a direct causal relation to the outcome variable (UTI). This promotes a potential elimination or modification of a risk factor, e.g. by reducing unnecessary use or improving management of indwelling urinary catheters in high-risk patients. Girard et al. thus identified the factors associated with HA-UTI in the general population of patients in geriatric hospitals, the most important of which were UTI in the six months before the study, and intermittent bladder catheterization [24]. We also found a strong correlation to previous UTIs in a general hospital population, which is likely to be associated with underlying urological pathologies in the patient. The small number of modifiable risk factors presently employed in our two predictive UTI models may constitute a limitation for the clinical use. However, preventive countermeasures may also involve a restriction on adding further UTI risk factors to the patients, e.g. the permanent use of an indwelling urinary catheter. Furthermore, countermeasures could also include implementation of augmented general infection control measures or more vigilance regarding early symptoms of urinary tract infection in high-risk patients. The initiation of rapid diagnostic and relevant treatment of UTI at an earlier clinical stage may prevent UTIs becoming a focal infection site for more serious infections such as blood stream infections.

Several studies have looked at developing prediction models with EMR data with the purpose of creating new analytic opportunities in diagnostics, prevention, treatment, and prediction of risk and outcome of various medical conditions including death [14]. Goldstein et al. published a systematic review of 108 published studies between 2009 and 2014 on the opportunities and challenges in developing risk prediction models using data from electronic medical records [25]. Only five of these 108 studies dealt with infections of which three focused on sepsis/bacteremia. No studies dealt specifically with predictions of risk of acquiring urinary tract infections during hospitalization. Goldstein et al. concluded that there in general was room for improvement in designing such studies. They specifically addressed that the advantage of the size of EMR data is not limited to number of patients but also to the access to a large number of potential predictor variables. They recommend that new studies should pay more attention

to the presence of missing data and informative presence. Both may lead to biased associations and impact prediction models.

Two studies have dealt specifically with the use of machine learning in relation to predicting the risk of urinary tract infections [12, 25]. Taylor et al. using a retrospective cohort study design examined algorithms for prediction of the outcome of a positive urine culture result ($>10^4$ colony forming units/mL) in a group of emergency department patients having clinical symptoms attributable to UTI and with urine samples taken for culture. In a single health-care center comprising four emergency departments they examined altogether 81,387 patients for analyses in an 80:20 split between a training and an evaluation cohort [26]. Among seven machine-learning methods XGBoost came best both in a full predictive model using 212 input variables (AUC = 0.904) and a smaller 10 input variable set up (AUC = 0.877). Interestingly, the XGBoost models predicted with a higher sensitivity an expected UTI diagnosis that a comparative health care provider judgement, 86.3% versus 41.7%, respectively. The full XGBoost model with 212 input variables performed only slightly better than the reduced model with 10 input variables. The authors state that a practical implementation of such a predictive model may thus offer the possibility of reducing empiric use of antibiotics.

More in line with our study, Hur et al. aimed to identify high-risk patients for catheter-associated UTI at an acute care hospital and to develop an automated risk assessment system for effective preventive measures against catheter-associated UTI [12]. Their study comprised 2150 patients of whom one fifth had a diagnosed catheter-associated UTI and the remaining patients were chosen as a non-UTI group, 1505 patients (70%) constituted the training set. Evaluation of their AutoRAS-UTI model was based on a few blood biochemistry test results and a short set of administrative data including length of stay indwelling urinary catheter application days. Their model had a high sensitivity of about 0.9 and a specificity of 0.88. By use of the model, they showed that the length of catheterization could potentially be reduced by reviewing the ongoing necessity of indwelling urinary catheters for high-risk patients and removing them as soon as possible. All hospitalized patients were on a daily basis divided into high- or low-risk groups and the risk assessment displayed on the patient summary screen with the high-risk group represented by a red color code.

Future studies of automated machine learning models for prediction of risk of acquiring UTI should not only try to refine machine learning algorithms but also look at their efficiency in routine use for promoting individual preventive measures against UTI developing in high-risk patients during hospitalization. We believe that our study can serve as a proof of concept, showing the potential use of predictive modelling in the battle against hospital acquired UTIs by creating daily awareness of an important clinical problem for individual patients. However, we do not regard the two models as standalone recipes for preventing specific hospital acquired infections.

Next step based on the web based reporting tool developed for our entry model is the introduction and evaluation of this system in one or more hospital departments with adequate software and hardware infrastructure. Routine implementation of the system for risk prediction of UTI should begin in wards with the highest rates of HA-UTI and followed up by monitoring the clinical effect of specific interventions for patients with a predicted high risk of acquiring a UTI. Importantly, aspects of clinical effectiveness such as patient outcomes and costs in terms of reduction in length of stay in the hospital should be measured. However, it is of key importance for implementation of the UTI prediction models in clinical practice to consider how well such a warning system will integrate into the daily workflow without undue increases of the alert burden on health care staff. Successful implementation of an early warning system however must go hand in hand with proper education of staff and

the awareness of the necessity of structural patient monitoring for prevention of hospital acquired infections.

## 5. Conclusion

We have shown that it is possible to create machine-learning models that can serve as early warning systems to predict patients in risk of acquiring a urinary tract infection during admission in the Region of Southern Denmark.

Both Models perform with a high ROC-index on the validation data set, Entry Model: 0.81 and HAI Model: 0.74 indicating a sufficient sensitivity and specificity, which potentially makes both models instrumental in individualized prevention of UTI in hospitalized patients. The selected machine-learning methodology is Decision Trees to ensure the most transparent results and understanding of the models for the health care staff.

We believe that our models is a promising tool for aiding health-care professionals in identifying high-risk patients by their UTI risk score and consider individual preventive measures to counteract development of UTIs during hospital stay. Further studies applying the entry model and HA-UTI model in a clinical practice are needed. We are preparing for a clinical evaluation of our entry model in a Department of emergency medicine. An automatic UTI alert button has been integrated in the patient flow management solution used at Vejle Hospital as an indicator on the summary screen of patients having a prediction score for acquiring UTI four times higher than the average risk for all patients admitted.

## Supporting information

**S1 Table. HAIR definitions of UTI.**
(PDF)

**S2 Table. Test characteristics of the UTI prediction models.**
(PDF)

**S3 Table. Variable importance for the HA-UTI model.**
(PDF)

**S1 Fig. Illustrating how model assessment chooses the least complex model with best model performance.**
(TIF)

**S2 Fig. Entry-model based on decision tree (example).**
(TIF)

**S3 Fig. Cumulative lift for HA-UTI model.**
(TIF)

**S4 Fig. Dashboards for operationalizing of the results (presenting an overview).**
(TIF)

**S5 Fig. Dashboards for operationalizing of the results (reporting individual patient).**
(TIF)

**S1 Appendix. Data from preliminary prospective study.**
(XLSX)

**S2 Appendix. SAS tools employed.**
(PDF)

## Acknowledgments

The authors are grateful to Department of documentation and management information, Health, Region of Southern Denmark for assistance in providing data from the regional data warehouse.

We thank Mike Johansson, Gudrun Bergman, and Lisa Kobæk from SAS institute Nordic Offices, and Pontus Nauclér, John Karlsson Valik, and Emil Thiman from Karolinska University Hospital in Stockholm for valuable and inspiring discussions.

## Author Contributions

**Conceptualization:** Jens Kjølseth Møller, Martin Sørensen, Christian Hardahl.

**Data curation:** Martin Sørensen, Christian Hardahl.

**Formal analysis:** Martin Sørensen.

**Funding acquisition:** Jens Kjølseth Møller, Christian Hardahl.

**Investigation:** Jens Kjølseth Møller, Martin Sørensen, Christian Hardahl.

**Methodology:** Jens Kjølseth Møller, Martin Sørensen, Christian Hardahl.

**Project administration:** Jens Kjølseth Møller, Christian Hardahl.

**Resources:** Jens Kjølseth Møller, Christian Hardahl.

**Software:** Martin Sørensen, Christian Hardahl.

**Supervision:** Jens Kjølseth Møller, Christian Hardahl.

**Validation:** Martin Sørensen, Christian Hardahl.

**Visualization:** Martin Sørensen.

**Writing – original draft:** Jens Kjølseth Møller.

**Writing – review & editing:** Jens Kjølseth Møller, Martin Sørensen, Christian Hardahl.

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
