## [Decision Letter · Decision Letter 0]

17 Jul 2020

PONE-D-20-17945

Prediction of risk of acquiring urinary tract infection during hospital stay based on Machine-learning: A retrospective cohort study

PLOS ONE

Dear Dr. Møller,

Thank you for submitting your manuscript to PLOS ONE. After careful consideration, we feel that it has merit but does not fully meet PLOS ONE’s publication criteria as it currently stands. Therefore, we invite you to submit a revised version of the manuscript that addresses the points raised during the review process.

We look forward to receiving your revised manuscript.

Kind regards,

Francesco Pappalardo, Ph.D.

Academic Editor

PLOS ONE

Journal Requirements:

Reviewers' comments:

Reviewer's Responses to Questions

**Comments to the Author**

1. Is the manuscript technically sound, and do the data support the conclusions?

Reviewer #1: Yes

2. Has the statistical analysis been performed appropriately and rigorously? 

Reviewer #1: Yes

3. Have the authors made all data underlying the findings in their manuscript fully available?

Reviewer #1: No

4. Is the manuscript presented in an intelligible fashion and written in standard English?

Reviewer #1: Yes

5. Review Comments to the Author

Reviewer #1: Healthcare associated infections (HAI), including urinary tract infections (HA-UTI) accounting for about 20-30% of all HAI’s, are a major health care burden. Case finding for the high-risk group of HA-UTI should be instrumental if sufficient sensitivity and specificity can be warranted. There have been previous efforts in a real-time early warning system for monitoring inpatient disease risk, for example PMID 31278734, which should be cited.

However, significant revision should be needed to make this work useful:

1. Data availability – the raw data matrix for each of the data mining modeling (training/testing) should be made available for download so that others can take a spin to replica their work or improve their work.

2. Machine learning approach – it is great that the authors tried multiple machine learning approaches. But the goal here is not to instruct the audience as how-to cherry pick methods but rather whether we can deliver a HA-UTI model for care management. These can be presented in the appendix if the readers have interest to review and learn.

3. Model and model features – the most effective model should be presented in detail:

a. How were the predictive features selected?

b. The predictive features should be presented with univariate analysis including but not limiting to odds ratio, p value etc.

c. Currently the feature table is hidden as supplementary. It should be presented in the main text. From the table, looks like the past UTI is the major driver.

d. Should the first time UTI and the repeated UTI modeling efforts be separated as two different learning processes?

e. Throughout the paper, there is no discussion of the underlying clinical causality in regard to the driving features? This is a major limitation. Not only we need to predict but also interpret.

Therefore, major revision is requested.

6. PLOS authors have the option to publish the peer review history of their article (what does this mean?). If published, this will include your full peer review and any attached files.

Reviewer #1: No

---

## [Author Response · Author response to Decision Letter 0]

14 Oct 2020

A rebuttal letter that responds to each point raised by the academic editor and reviewer(s) have been upload labeled 'Response to Reviewers'.

A marked-up copy of our manuscript that highlights changes made to the original version is uploaded as a separate file labeled 'Revised Manuscript with Track Changes'.

An unmarked version of our revised paper without tracked changes is uploaded as a separate file labeled 'Manuscript'.

---

## [Decision Letter · Decision Letter 1]

19 Jan 2021

PONE-D-20-17945R1

Prediction of risk of acquiring urinary tract infection during hospital stay based on Machine-learning: A retrospective cohort study

PLOS ONE

Dear Dr. Møller,

Thank you for submitting your manuscript to PLOS ONE. After careful consideration, we feel that it has merit but does not fully meet PLOS ONE’s publication criteria as it currently stands. Therefore, we invite you to submit a revised version of the manuscript that addresses the points raised during the review process.

We look forward to receiving your revised manuscript.

Kind regards,

Francesco Pappalardo, Ph.D.

Academic Editor

PLOS ONE

Reviewers' comments:

Reviewer's Responses to Questions

**Comments to the Author**

1. If the authors have adequately addressed your comments raised in a previous round of review and you feel that this manuscript is now acceptable for publication, you may indicate that here to bypass the “Comments to the Author” section, enter your conflict of interest statement in the “Confidential to Editor” section, and submit your "Accept" recommendation.

Reviewer #2: All comments have been addressed

2. Is the manuscript technically sound, and do the data support the conclusions?

Reviewer #2: Yes

3. Has the statistical analysis been performed appropriately and rigorously? 

Reviewer #2: Yes

4. Have the authors made all data underlying the findings in their manuscript fully available?

Reviewer #2: Yes

5. Is the manuscript presented in an intelligible fashion and written in standard English?

Reviewer #2: Yes

6. Review Comments to the Author

Reviewer #2: The manuscript looks well organized and professionally presented. The conclusions sound interesting, and appropriate for the clinical context. Written English is correct, even if some sentences are wordy and could be shorted to enhance the readability.

Minor revision:

1. Section 2.3 (Data set creation and definitions), line 125. Please, the authors should introduce a reference to the previous work they are reporting.

2. Section 2.3 (Data set creation and definitions), line 139. The work is based on a period ranged from January 2017 to May 2018. Please, the authors should express their opinion about the length of the time range. Do the authors have the feeling of whether it is a short period? In that case, how they think to approach the solution in their next research?

3. Section 2.5 (Predictive modeling approach), line 229. Please, the authors should explain which “stopping criteria” they used.

4. Section 4 (Discussion), line 335-336. Please, can the authors give an estimation time to predict for a new-entry patient?

5. Section 5 (Conclusion), line 451-452. Please, can the authors spend more words to detail future works? How they plan to continue their research project?

7. PLOS authors have the option to publish the peer review history of their article (what does this mean?). If published, this will include your full peer review and any attached files.

Reviewer #2: No

---

## [Author Response · Author response to Decision Letter 1]

26 Feb 2021

Response to Reviewers (PONE-D-20-17945R1)

Please find our response to the comments of the reviewers (shaded grey) below. 

Comments to the Author (on previous round of review)

1. If the authors have adequately addressed your comments raised in a previous round of review and you feel that this manuscript is now acceptable for publication, you may indicate that here to bypass the “Comments to the Author” section, enter your conflict of interest statement in the “Confidential to Editor” section, and submit your "Accept" recommendation.

Reviewer #2: All comments have been addressed

Response: We are happy to know that Reviewer #2 finds that all comments raised by Reviewer #1 in a previous round of review have been addressed.

6. Review Comments to the Author

Reviewer #2: The manuscript looks well organized and professionally presented. The conclusions sound interesting, and appropriate for the clinical context. Written English is correct, even if some sentences are wordy and could be shorted to enhance the readability.

Response: We thank the reviewer for the positive comments. We agree that some sentences could be shortened or enhanced by breaking up into two sentences. Examples of redundant text have also been removed. Changes are marked in yellow in the marked-up copy of the revised manuscript.

Page 4, lines 70-73.

Page 18, lines 362-365.

Page 18, lines 369-372

Page 19/20, lines 403-407

Page 20, lines 414-425 and lines 431-437

Page 21, lines 447-451

Page 22, lines 471-474

Minor corrections of the main text have been made (these changes have also been marked in yellow in the revised manuscript).

Page 17, lines 346. Figure changed from 75% to 72% (185/256).

Page 18, line 360. ‘about 75% of patients’ changed correspondingly to ‘close to three quarters of patients’.

Minor revision:

1. Section 2.3 (Data set creation and definitions), line 125. Please, the authors should introduce a reference to the previous work they are reporting.

Response:

Three additional literature references are included in Section 2.3 to introduce and support previous work on computer based algorithms, trigger tools, and text mining:

Gerdes LU, Hardahl C. Text mining electronic health records to identify hospital adverse events. Text mining electronic health records to identify hospital adverse events. Stud Health Technol Inform. 2013;192:1145. doi:10.3233/978-1-61499-289-9-1145.

Redder J. Surveillance of Hospital-Acquired Infections with Special Focus on Urinary Tract Infections: Development of an Automated Monitoring System. Ph.D. thesis, Syddansk Universitet. Institut for Regional Sundhedsforskning, 2016.

Mevik K, Hansen TE, Deilkås EC, Ringdal AM, Vonen B. Is a modified Global Trigger Tool method using automatic trigger identification valid when measuring adverse events? Int J Qual Health Care. 2019 Aug 1;31(7):535-540. doi: 10.1093/intqhc/mzy210.

2. Section 2.3 (Data set creation and definitions), line 139. The work is based on a period ranged from January 2017 to May 2018. Please, the authors should express their opinion about the length of the time range. Do the authors have the feeling of whether it is a short period? In that case, how they think to approach the solution in their next research?

Response: We find that more than one year of data is sufficient. We have made the following addition to the text on page 7, lines 148-150 to underline this opinion:

Seasonal changes of UTI infection patterns may occur but the present use of more than one year of data should be sufficient to generalize for scoring new cases. From a modelling perspective, we are able to create models that perform well on new data.

3. Section 2.5 (Predictive modeling approach), line 229. Please, the authors should explain which “stopping criteria” they used.

Response: We have explained in more detail the “stopping criteria” on page 12, lines 247-255:

The training process stops when one of a set of model specifications / stopping criteria has been met. The criteria used are the following: Neural Network, (number of hidden layers: 1, number of hidden units: 10, max. iterations: 50, max. time: 4 hours); Gradient Boosting, (number of iterations: 50, max. branch: 2, max. depth: 3, min. categorical size: 5, min. leaf fraction: 0.001); Regression, model selection based on the stepwise selection method; Decision Tree, (max. tree depth: 6, min. leaf size: 5, min. categorical size: 5, max. branch: 2); Decision Tree 3 Way Split, (same criteria as the Decision Tree Model, but with max. branch: 3). The primary assessment metric used is average squared error on validation data.

4. Section 4 (Discussion), line 335-336. Please, can the authors give an estimation time to predict for a new-entry patient?

Response: We agree that this is important information. The following text has been added on page 19, lines 382-387 to clarify this issue:

With the current data warehouse setup, the Entry Model is updated with new admitted patients hourly. Estimated time from a patient is registered in the database to a prediction is returned to the clinical system is only a matter of seconds for one patient. Depending on the right integrations to the source system it would be possible to do this in near real-time. Changes in classification codes of sample material and pathogens cultured should currently be reflected in the UTI algorithms of HAIR [6, 7].

5. Section 5 (Conclusion), line 451-452. Please, can the authors spend more words to detail future works? How they plan to continue their research project?

Response: Future work is part of the discussion on Page 22, from line 475. We agree that part of this discussion explaining planned future studies in more detail is more suitable placed in the conclusion. Therefore, we have moved the following section from the last part of the discussion on Page 23 to the Conclusion on page 24, lines 514-518:

We are preparing for a clinical evaluation of our entry model in a Department of emergency medicine. An automatic UTI alert button has been integrated in the patient flow management solution used at Vejle Hospital as an indicator on the summary screen of patients having a prediction score for acquiring UTI four times higher than the average risk for all patients admitted.

---

## [Editor Report · Decision Letter 2]

3 Mar 2021

Prediction of risk of acquiring urinary tract infection during hospital stay based on Machine-learning: A retrospective cohort study

PONE-D-20-17945R2

Dear Dr. Møller,

We’re pleased to inform you that your manuscript has been judged scientifically suitable for publication and will be formally accepted for publication once it meets all outstanding technical requirements.

Kind regards,

Francesco Pappalardo, Ph.D.

Academic Editor

PLOS ONE
---

## [Editor Report · Acceptance letter]

18 Mar 2021

PONE-D-20-17945R2 

Prediction of risk of acquiring urinary tract infection during hospital stay based on Machine-learning: A retrospective cohort study 

Dear Dr. Møller:

I'm pleased to inform you that your manuscript has been deemed suitable for publication in PLOS ONE. Congratulations! Your manuscript is now with our production department. 

Kind regards, 

on behalf of

Prof. Francesco Pappalardo 

Academic Editor

PLOS ONE